# Can Microsaccades Be Used for Biometrics?

**DOI:** 10.3390/s23010089

**Published:** 2022-12-22

**Authors:** Kiril Alexiev, Teodor Vakarelski

**Affiliations:** Institute of Information and Communication Technologies, Bulgarian Academy of Science, 25A Acad. G. Bonchev Str., 1113 Sofia, Bulgaria

**Keywords:** eye tracking, fixational eye movements feature selection, microsaccade, biometrics

## Abstract

Human eyes are in constant motion. Even when we fix our gaze on a certain point, our eyes continue to move. When looking at a point, scientists have distinguished three different fixational eye movements (FEM)—microsaccades, drift and tremor. The main goal of this paper is to investigate one of these FEMs—microsaccades—as a source of information for biometric analysis. The paper argues why microsaccades are preferred for biometric analysis over the other two fixational eye movements. The process of microsaccades’ extraction is described. Thirteen parameters are defined for microsaccade analysis, and their derivation is given. A gradient algorithm was used to solve the biometric problem. An assessment of the weights of the different pairs of parameters in solving the biometric task was made.

## 1. Introduction

Human eyes are in constant motion. Even when we fix our gaze on a certain point, our eyes continue to move [1,2,3]. Scientists have distinguished three different FEMs—microsaccades, tremor and drift [4,5].

Microsaccades are the fastest eye movements compared to the other two movements. Unlike macrosaccades [6], which are used to direct the eyes to different objects in a scene, microsaccades have a significantly smaller amplitude on the order of 12 arcmin [7,8], and their purpose is still not fully understood [9]. Some scientists believe that microsaccades are simply impulse noise in the system; others suggest that the microsaccades have the function of repositioning the image on the fovea during fixation periods [10]. There are also opinions that the microsaccades have a compensatory function to stabilize the image on the fovea during head and body movements, as their control comes from the visual and vestibular part of the brain [11]. In [12,13], the researchers claim that the microsaccades are used for gaze fine-tuning and that they play an important role in attentional concentration.

A tremor is a movement of the eye of the highest frequency in fixation mode. A microtremor is an involuntary eye movement with a relatively small amplitude. Scientists define the frequency range of a tremor usually in the range of 30 to 110 Hz. In the present work, we have assumed that the tremor range is 70–103 Hz [14].

A drift is a slow and smooth movement of the eyes in fixation mode. The frequency of ocular deviations is significantly lower than that of microtremors and is generally considered to be in the range 0–40 Hz, and the amplitude of change is about 1.5′–4′ with an average speed of about 4′/s [15]. There are different hypotheses about the functional purpose of drifts. One of these hypotheses suggests that eye drift helps to obtain higher resolution when observing stationary objects [16,17]. Another paper suggested that fixation-mode eye drift is related to the encoding and processing of visual information in space and time [18]. Recently, the authors of [19,20] showed that drift is inversely correlated with the head/body movements with some delay in time. This means that the eye drift’s inverse correspondence to these movements gives us information about the velocities and amplitudes of body and head micro movements.

During fixations, details of the image are perceived, which is an essential part of the cognitive process. Because eye fixation tremors are of small amplitude and mainly attributed to noise in the system [21], and drift plays a largely compensatory function, our main interest in this article is directed at microsaccades, regardless of their unclear origin and physiological purpose or perhaps precisely because of these. The article proposes an analysis of microsaccades in a scenario of voluntary gaze concentration on a point. The choice of this scenario is determined by our desire to make the trials as independent as possible from the content of the solved cognitive task. The dependence of microsaccades on the content of the specific cognitive task is not the subject of investigation in the present paper. It is of interest to find out if there is anything common among the microsaccades of the observed people in the case of the described above simplified fixative task. Another question is: how different are the microsaccades of the observed individuals? The pilot study of one, albeit small, group of people reported in this article shows the existence of significant differences in the microsaccades of individuals. At the same time, it turned out that the microsaccades in separate experiments with the same person have much in common and are undoubtedly quite similar. In order to parameterize similarity/dissimilarity, it is necessary to find appropriate characteristics of the microsaccades. A set of microsaccade features is proposed and analyzed in this article. The importance of each of the selected features and the pairs of features for the needs of biometrics were investigated.

The paper is organized as follows. The following section reveals the current state of research in the field of using eye tracking data for biometrics needs. In the third section the proposed method is described. It includes preprocessing of the eye tracker data stream, microsaccade segmentation, parameterization, definitions of distance functions and weight computation. The fourth section presents experimental design, algorithm description and received results. The last chapter summarizes the results and outlines the most important conclusions and directions for further research.

## 2. Related Work

The question discussed in the previous section about whether microsaccades can be used for biometrics purposes is part of the more general problem of using eye movements in human recognition. The use of computer technology to extract the unique traits (whether physical or behavioral) of a person in order to verify or identify their identity is at the core of biometrics.

Physically based biometric modalities can be thought of as a snapshot of one of the physical parts of our bodies. For the needs of biometrics, one or more such photos can be used, which are based on the corresponding material carrier (part of our body).

Biometric modalities that are based on behavior usually need much more information about the dynamics and statistical characteristics of any of the processes in our bodies. Even though they are caused by a particular material medium, this information cannot be found directly in the parts of our body—i.e., it is without a material component.

The beginning of the use of eye movement as a modality in so-called soft biometrics or non-intrusive biometrics began in 2004 with the work of Kasprowski and Ober [22]. In the following year, Bednarik et al. [23] investigated static and dynamic eye parameters and eye movement for biometrics needs. Later, in 2006, Silver and Biggs [24] investigated whether eye movements and text writing could be used for human identification. In the period between 2010 and 2015, numerous articles were published in which different approaches were proposed for using eye movements in biometrics [25,26,27,28,29,30,31,32,33,34,35,36]. One distinguishing feature of the most of them is that statistical characteristics of saccadic movements or eye fixations are used; the amplitude, time delay or speed of saccades, and in some cases even accelerations, are studied. In Komogortsev’s approach, a mathematical model of the oculomotor group of muscles (oculomotor plant characteristics) was utilized. The model was parameterized by using the obtained data, and the identification of a person is sought in the parameter space. In 2012 the Eye Movement Verification and Identification Competition (EMVIC) at BTAS 2012 was held [37]. The competition used publicly available eye movement databases. The data were divided into two groups—for training and for testing. After that, two more competitions, EMVIC 2014 and the BioEye 2015 competition, were organized [38]. They were dominated by methods using both the personal anatomical features of the eyes and the behavior of the eyes when solving a specific visual task. However, a common characteristic of these approaches is that biometric identification is performed using information derived primarily from fixations and saccades. This information usually includes the number and duration of fixation points, various saccade parameters such as average saccade amplitudes, average saccade velocities, average saccade peak velocities, the velocity waveform, eye movement parameters when viewing a given scenario such as scan path length, scan path area, scan path inflections, regions of interest, etc.

In some articles [32] other non-fixational eye movements such as nystagmus, smooth pursuit and vestibulo-ocular reflex eye movements were considered to be rejected from further research. Any information contained in the fixational micromovements was also ignored.

In the last years new features have been introduced to define vectors used in classification based on nonlinear time series analysis [39]. Three ranges of eye movement were included: saccadic latency—the time it takes the brain to respond to a change in stimulus position—saccade and fixation. The analysis is performed on each segment of 100 points, for which 23 features are calculated. Yin et al. [40] proposed a neural network for extracting eye movement information and constructing a saccade propagation map. Only large eye movements are considered, and the classification is performed based on the extracted information. In [41], eye movement data from 39 subjects in a real-world orientation experiment were processed. The selected eye movement characteristics included pupillary response, fixation point density, scene semantics at fixation points and several saccade characteristics. Yoo et al. [42] also analyzed eye movement behavior, emphasizing the salience of visual stimuli (again, attention is paid to the semantics of the scene). A low-pass filter was used to remove fixation movements that were considered noise in the system.

What is new that has appeared in recent years is the combination of several modalities, for example the combination of eye behavior and iris structure or the anatomical features of the eyes with their behavior/strategy when observing a given scene.

Incorporating the analysis of eye behavior when viewing a scene creates two serious problems for the use of eye data in biometrics. The first of them is related to the need to record and process a sufficiently long (in time) record of eye activity. Due to the fact that saccades and fixations are low-frequency events, the recording time is usually well over a few seconds. The second problem is related to the fact that usually eye behavior is determined by the type of visual task being solved and the specific scenario. This makes the analysis task-dependent—an undesirable feature for the needs of biometrics.

The additional information contained in high-frequency micromotions has the potential to significantly speed up identification. Microsaccades are one of these movements in fixation mode. Unlike some other fixational eye movements, microsaccades are performed in a coordinated manner by both eyes. From a biological point of view, this means that a synchronizing control signal is generated for their execution. They could be used for image stabilization and anti-fading [43], although it has been already shown that microsaccades are not absolutely necessary to prevent perceptual fading and to control fixation position. Rather, microsaccades and drift work together to realize these functions. Despite the ambiguous origin of microsaccades, it is clear that they play an important biological and cognitive role, and this makes them a suitable tool to be used for biometric purposes. To our knowledge, this is the first work devoted to the use of microsaccades in biometrics, which could enable an entirely new class of soft biometrics based on fixational eye movements to be explored.

## 3. Proposed Method

Eye movement information is gathered by using an eye tracker. To obtain normalized information from sensors, it is necessary to calibrate them at the beginning of the experiment. In our study only the eye movement information in fixation mode is used. In order to maximally decorrelate the data from solving a specific cognitive task, the attention of the observed person is attracted by the appearance of a dot (single) on a screen in a darkened room. The eye tracker generates a data stream with the coordinates of the gaze. The sections of the data stream in which a person’s gaze is directed at the point must be found [44]. These sections are fetched and stored for further processing. Various types of filters are used to remove high-frequency noise from the stored signals [45,46]. Then, the three different types of eye movements in fixation mode are separated. All the processes described up to now are called eye tracker data preprocessing. The separated microsaccade movements are subjected to the following processing: feature extraction, weighting of individual features by error function optimization and human classification.

### 3.1. ET Data Preprocessing

Here, a short explanation of the preprocessing steps for the ET data is given and shown schematically in Figure 1. A detailed description can be found in [20].

When conducting the experiments involving holding the gaze on a single point, we obtain measurements from the ET with information about the small eye movements during fixation. The three types of movements during fixation (microsaccades, microtremors and drifts) differ significantly in their frequency and amplitude characteristics, and this gives us reason to accept the hypothesis of their additivity. We also assume that the measurement process is accompanied by the presence of Gaussian distributed additive noise [47].

The processing of the obtained measurements starts with reducing the influence of noise (denoising/smoothing). The filtering is performed by using two different algorithms: spectral filtering and Kalman filtering. The spectral filtering algorithm is the simplest one possible. It applies prior knowledge about the maximum frequency of eye movements (about 104 Hz). After a Fourier transform, all amplitudes in the frequency domain above 104 Hz are zeroed, and then the reverse Fourier transform returns the filtered signal in the original time domain. The second algorithm—Kalman filtering—is built on a 2D model of eye movement, containing a state vector (velocity and position of the eye in horizontal and vertical directions/axes) and a transition matrix [48], which are describing the physical model of eye movement. The error covariance matrices of the state and measurement vectors are calculated according to expressions in [20] or replaced with the values estimated in [47]. The noise filtration procedure ends with power supply harmonics rejection (at 50 Hz and 100 Hz; others were already removed with previous filtering).

The second step consists of microsaccade detection. Three different algorithms were proposed in [20]. Two of them could be regarded as enhanced versions of the well-known classical Engbert and Kliegl algorithm [49,50] with a different window size and adaptive threshold. The third algorithm is statistical. It looks for a change in statistical parameters of ET data by chi-square distribution with 10 degrees of freedom (a window from 11 measurements, which corresponds to 100 Hz). As a result of the execution of this step, we obtain the fragments of the trajectory of the eyes in the state of fixation, which are classified as microsaccades.

The third and final step in the processing of the obtained ET measurements involves the detection and separation of the other two eye movements in the fixation mode: drift and tremor. Because of their unique frequency characteristics [51], they can be easily extracted from the frequency domain.

The ET data preprocessing filters and divides the noisy ET data into three classes of FEM: microsaccades, tremors and drifts. It is important to remark that the classes of microsaccades and drifts consist of multiple single microsaccades and single drift movements, often shifted significantly according to each other.

### 3.2. Motivation for Microsaccade Analysis and Microsaccade Parameterization

Some of the obtained experimental results are shown in Figure 2. Eye movements are visualized exclusively in fixation mode without noise (filtered in advance) or parasitic eye movements between individual fixation points (also removed). Each row corresponds to the eye movements of the same person at 4 or 5 fixation points. The table contains data collected for 8 people. Typically eye movements of greater amplitude are microsaccades. Thanks to the built-in analytical mechanisms in our brain, it is easy to see similarity in the microsaccades of the same person and significant differences between the microsaccades of different participants in the experiments.

In the next figure (Figure 3), only the fixation microsaccades are visualized. Their graphical representation is not normalized, and the presence of relatively large offsets greatly rescales the images along one or both axes simultaneously to a different scale. However, the dependencies in the eye movements of the same person found in complex fixational eye movements are even more clearly visible.

Let us denote by S the set of extracted microsaccades on one fixative point and call it a segment. We are interesting in similarity between segments of a person and dissimilarity of segments belonging to two different persons. A quantitative measure of similarity is very important. It could be used for person identification, detection of drug use, estimation of mental health and in many other applications.

For each application a predefined criterion C for every segment S or a couple of segments ($S_{1},S_{2}$) should be provided. For example, $C_{1}$: “Do $S_{1}$ and $S_{2}$ belong to one and the same person?” should be used in people identification systems. Or $C_{2}$: “Does S belong to a person, who has used drugs or alcohol?” could be used for drug detection systems [52]. Or $C_{3}$: “Does S belong to a person with a good mental health?” could be used for mental health estimation systems. Or $C_{4}$: “Does S belong to a person with an aggressive temper?” could be used for the psychological classification of persons. Or $C_{5}$: “Does S belong to a person, who is an accurate shooter?” could be used for sport estimation systems. Or $C_{6}$: “Does S belong to a person with healthy eye oculomotor group” could be used for a medical eye health system. Each criterion C should be carefully selected.

From the above information, it follows that the quantitative analysis of the similarity/difference among two segments will open up opportunities for multiple applications in a variety of fields. In order to implement a quantitative analysis of proximity, it is necessary to describe the microsaccades in an appropriate way. It is necessary to define some features that meet the following conditions [53]: universality (every segment possesses the chosen characteristics); uniqueness (no two persons have similar segments in terms of chosen characteristics); permanence (the chosen characteristics are invariant in time); and collectability (the features are measurable in a quantitative way).

Next, 13 parameters are proposed to describe microsaccades. A class of functions is also defined, within which the optimal one is sought for person identification based on fixational microsaccades. The resulting solution shows good discrimination characteristics even on the basis of the limited data available in the pilot study.

### 3.3. Parameterization of the Microsaccades

The duration of holding the gaze at one point in the selected scenario is 15 s. All microsaccades at this fixation were collected into a segment. The parameterization is performed within a single microsaccade, which means that the microsaccade features are defined on a part of the points describing the trajectory of the gaze in a segment. For this purpose, we describe each microsaccade with an ordered set of measurements $Z_{k+1,k+n}=\left\{z_{k+1},z_{k+2},\ldots z_{k+n}\right\}$, where $z_{k+1}$ and $z_{k+n}$ are the first and the last points of a considered microsaccade, respectively, and n is its length. Each point $z_{k}$ is defined by an ordered pair of coordinates $\left(x_{k},y_{k}\right)$ corresponding to the position of the sight at the specific time moment k.

A brief description of the selected parameters follows below.


**Duration**


The duration of a microsaccade, measured in milliseconds, corresponds to the number of sample points of the considered microsaccade. The frequency of the eye tracker is 1000 Hz, so a microsaccade’s duration is determined by the number of measurements in the microsaccade sample minus one. For example, a microsaccade with a length of 50 measurements corresponds to a duration of 49 ms.


**Height**


Let us denote the Euclidean distance between measurements
$z_{i}$ and
$z_{j}$ by
$d_{i,j}=\sqrt{{\left(x_{i}-x_{j}\right)}^{2}+{\left(y_{i}-y_{j}\right)}^{2}}$. Let the first and the last measurements of a microsaccade be respectively denoted by
$z_{k+1}$ and
$z_{k+n}$. The height is defined as the maximum distance between the line determined by
$z_{k+1}$ and
$z_{k+n}$ and the measurement
$z_{k+i},\;1<i<n$:
$$H=\left\{\begin{array}{c} \underset{1<i<n}{max}\frac{2\sqrt{p\left(p-d_{k+1,k+i}\right)\left(p-d_{k+n,k+i}\right)\left(p-d_{k+1,k+n}\right)}}{d_{k+1,k+n}},\;if\;d_{k+1,k+n}>0\;\\ \underset{1<i<k}{max}d_{k+1,k+i},\;if\;d_{k+1,k+n}=0\; \end{array}\right.$$
where $p=\frac{d_{k+1,k+i}+d_{k+n,k+i}+d_{k+1,k+n}}{2}$.


**Area**


The area of each microsaccade is calculated with a discretization of Green’s formula:$$S=\overset{n-1}{\underset{i=1}{\sum }}\frac{\left(x_{i+1}+x_{i}\right)}{2}\left(y_{i+1}-y_{i}\right)=\frac{1}{2}\overset{n-1}{\underset{i=1}{\sum }}\left(x_{i}y_{i+1}-y_{i}x_{i+1}\right)$$

The expression above is obtained only for a non-intersecting closed trajectory of a microsaccade.

When the trajectory of a microsaccade is self-crossing, it can be divided to smaller non-self-intersecting polygons with respect to the intersection points. The total area should be calculated as the sum of all the areas of these smaller polygons. The intersection points can be found using their properties. For every self-intersecting line composed of discrete points, there are at least two consecutive pairs of points AB and CD for which there exists a point incident to both line segments (AB and CD). If $A=z_{k};B=z_{k+1};C=z_{l};D=z_{l+1}$; and l≠k−1,k,k+1, then:$$\lambda z_{k}+\left(1-\lambda \right)z_{k+1}=\mu z_{l}+\left(1-\mu \right)z_{l+1},$$
where $\lambda \in \left[0;1\right]$ and $\mu \in \left[0;1\right]$. If the equation is rewritten for coordinate components it looks like:$$\left\{\begin{array}{c} \lambda x_{k}+\left(1-\lambda \right)x_{k+1}=\mu x_{l}+\left(1-\mu \right)x_{l+1}\\ \lambda y_{k}+\left(1-\lambda \right)y_{k+1}=\mu y_{l}+\left(1-\mu \right)y_{l+1} \end{array}\right.$$
or
$$\left(\begin{array}{cc} x_{k}-x_{k+1} & -x_{l}+x_{l+1}\\ y_{k}-y_{k+1} & -y_{l}+y_{l+1} \end{array}\right)\left(\begin{array}{c} \lambda \\ \mu \end{array}\right)=\left(\begin{array}{c} x_{l+1}-x_{k+1}\\ y_{l+1}-y_{k+1} \end{array}\right)$$

If the matrix is invertible and the both of the components of the solution are in [0;1], a self-crossing is observed, and the border points have the indexes k,k+1,l,l+1. If the matrix is invertible and at least one of the components of the solution is not in [0;1], then a self-crossing is not observed. If the matrix is not invertible, the considered pairs of points are parallel.


**Sharpness**


The sharpness is defined as the ratio: $S_{r}=\frac{H}{S}$.


**Base Length**


The base length is defined as the distance between the first and the last measurements of a microsaccade—$d_{k+1,k+n}$.


**Double Microsaccade**


A double microsaccade is defined as microsaccade whose start and end points are close with respect to the whole length of the microsaccade. Some microsaccades are double, while others are mono. There are cases when a microsaccade is hard to classify as mono or double. A quantitative definition is introduced in order to distinguish mono from double microsaccades—the ratio of the base length to the microsaccade duration.


**Quantitative definition of a double microsaccade**


Double microsaccades are more widely discussed in the literature. Usually, the main cause of a mono microsaccade is an involuntary head movement. A head movement, however, cannot create a double microsaccade. The start and end points of double microsaccades are very close to each other with respect to the microsaccade duration time and also to the distance between these points. This explicates why a double microsaccade is not regarded as a compensatory mechanism of a head movement.

Due to the different origins of mono and double microsaccades, it is important to distinguish them and process them separately in the algorithm. The chosen threshold is set to 0.01 (the ratio of the base length to the microsaccade duration). If the ratio is higher than the threshold, the microsaccade is considered mono. If the ratio is lower than the threshold, the microsaccade is considered double.


**Average speed**


Let the measurements from a microsaccade be denoted by $z_{1},z_{2},\cdots ,z_{n}$, where n is the number of all points in the microsaccade. The corresponding velocities $v_{1},v_{2},\cdots ,v_{n-1}$ are defined as $v_{i}=d_{i,\;i+1}$. The average speed is:$$v_{av}=\frac{\left(v_{1}+v_{2}+\cdots +v_{n-1}\right)}{n-1}$$


**Maximal Speed**


The definition for maximal speed is more complex. Eye velocity can be extremely high at certain points—partly due to the high angular velocity of the eye movement in a microsaccade and partly due to insufficiently well-removed sensor noise, head movements, body tremors or a combination of these. The differentiation process amplifies the influence of small errors and noise, and therefore the determination of the maximum speed is performed in a window in which the noise is further smoothed to some extent. Two windows with different lengths are proposed:


**Window with length of 11 velocities**


The average speed at i is calculated by averaging 11 sequential velocities:$$v_{av11,i}=\frac{v_{i-5}+\cdots +v_{i+5}}{11}$$

The maximal speed is determined by:$$v_{max11}=\underset{5<i<n-5}{max}\left\{v_{av11,i}\right\}$$


**Window with length of 21 velocities**


The same algorithm is used, with the only difference being that the number of averaged velocities is set to 21:$$v_{max21}=\underset{10<i<n-10}{max}\left\{v_{av21,i}\right\}$$


**Average pseudo acceleration**


The pseudo acceleration is defined as the first derivative of the magnitude of the speed. The following discrete formula is used: $a_{i}=v_{i+1}-v_{i}$. The average amount is estimated with the formula:$$A_{av}=\frac{a_{1}+\ldots +a_{\left(n-2\right)}}{n-2}$$


**Averaged maximal pseudo acceleration (10 points of approximation)**


A different algorithm is used to determine the maximum acceleration in comparison with the maximal speed calculation. Once the accelerations are found, they are ranked by magnitude, and the top 10 are used to find the largest acceleration by averaging them.


**Maximal diameter**


Given the measurements of a microsaccade $z_{1},\cdots ,z_{n}$ the distances ${\left\{d_{i,j}\right\}}_{1\leq i\leq n,1\leq j\leq n,\;i\neq j}$ are calculated, where $d_{i,j}$ is the distance between the points $z_{i}$ and $z_{j}$. The maximal diameter $D_{max}$ is the maximum value in the set of the distances:$$D_{max}=\underset{1\leq i\leq n,1\leq j\leq n,\;i\neq j}{max}d_{i,j}$$

### 3.4. Distance Functions

The problem of constructing a function that describes or evaluates how similar two microsaccades are is cardinal to obtaining an answer to the main question of this study: is it possible to identify individuals by using their fixational microsaccades? It is also interesting to understand how important each of the selected parameters is to the identification process.

Furthermore, several distance functions are proposed, which include the selected parameters and their weighting factors. The selected parameters describe certain essential characteristics of a microsaccade. The products of all possible pairs of microsaccade parameters are also considered. The combination of two parameters adds more complexity to the study because it combines parameters describing radically different properties of microsaccades. To each parameter or product of two parameters is assigned a weighting factor depending on its contribution to the final successful identification.

The distance functions are defined for the simplest case—studying the proximity of two microsaccades—for two segments (a set of random numbers of microsaccades) and for several fixation points (for observed individuals or for two sets of segments).

#### 3.4.1. Linear Distance Function

Let $A=Z_{k+1,k+n}^{A}$ and $B=Z_{m+1,m+l}^{B}$ be two microsaccades, represented by k and l measurements, respectively. Let $\left\{P_{1}^{A},P_{2}^{A},\ldots ,P_{13}^{A}\right\}$ and $\left\{P_{1}^{B},P_{2}^{B},\ldots ,P_{13}^{B}\right\}$ be the calculated corresponding parameters of these two microsaccades. For example $P_{1}^{A}$ is the duration of the microsaccade A, and $P_{2}^{B}$ is the height of the microsaccade B.

A function ρ:M×M→R is defined as a weighted sum of features, where M is the set of all observed saccades:$$\rho \left(A,B\right)=\sum_{i=1}^{13}w_{i}{\left(P_{i}^{A}-P_{i}^{B}\right)}^{2},$$
where $w_{1},\cdots ,w_{13}$ are the weights for each parameter. This function defines how “close” two microsaccades are. The weights in the function indicate the importance of the corresponding parameters in determining the similarity of the considered two saccades.

#### 3.4.2. Quasilinear Distance Function

The second proposed function takes into account both the impact of single parameters and all possible pairs of them.
$$\rho_{ql}\left(A,B\right)=\sum_{i=1}^{13}w_{i}{\left(P_{i}^{A}-P_{i}^{B}\right)}^{2}+\sum_{i=1}^{12}\sum_{j=i+1}^{13}w_{i,j}\left|P_{i}^{A}-P_{i}^{B}\right|\left|P_{j}^{A}-P_{j}^{B}\right|,$$
where $w_{1},\cdots ,w_{13},w_{1,2},\cdots ,w_{12,13}$ are the weights of each parameter and each combination of two parameters. A useful simplification is the notation ${\left\{w_{i}\right\}}_{i=1}^{N}$, where all weights are indexed in a series, and:$$N=n+C_{n}^{2}=\frac{n\left(n+1\right)}{2}=91$$

Building this vector of the weights simplifies further formulas and definitions and brings a generality to the description.

The following three properties are valid for both the linear and quasilinear cases:(1)$\rho \left(A,B\right)=0\Leftrightarrow x=y$(2)$\rho \left(A,B\right)=\rho \left(B,A\right)$(3)$\rho \left(A,B\right)\leq \rho \left(A,C\right)+\rho \left(C,B\right)$

Due to the fulfillment of the above three conditions, the proposed linear and quasilinear functions play the role of distances. All further algorithms are developed for distance functions.

#### 3.4.3. Segment Distance

Every individual is observing five static points. All microsaccades during the observation of one fixation point form one segment. The number of microsaccades is random, and it changes even in the records of the same person. Therefore, it is necessary to determine the distance between two segments containing in general a different number of microsaccades. Let us denote the compared segments by $T_{1}$ and $T_{2}$. We will set three types of distances. The first one of them is:$$d_{1}\left(T_{1},T_{2}\right)=\underset{A\in T_{1},B\in T_{2}}{min}\left(\rho \left(A,B\right)\right),$$
where A and B are referring to all microsaccades from the sets/segments $T_{1}$ and $T_{2}$, respectively.

The defined function is fulfilling the first and the second distance properties, but the third property is not fulfilled—the triangle inequality. This function is not a distance function but an approximation of one. This function has a very useful property: it is automatically removing outliers that would abnormally increase the distance between two segments.

The second definition of the distance between two segments is defined to satisfy all three distance conditions. For this purpose, we find the centers of mass of the parameters describing all microsaccades in one segment. Let us denote the centers of mass of the parameters for the segments $T_{1}$ and $T_{2}$ by the vectors ${\bar{t}}_{1}$ and ${\bar{t}}_{2}$, respectively. The vectors ${\bar{t}}_{1}$ and ${\bar{t}}_{2}$ are lying in the same 13-dimensional parameter space as the microsaccade parameters of $T_{1}$ and $T_{2}$. A proper distance function could be defined as:$$d_{2}\left(T_{1},T_{2}\right)=\rho \left({\bar{t}}_{1},{\bar{t}}_{2}\right)$$

An additional step is proposed in the algorithm to preserve the outlier-removing property. Let $T_{1}^{\prime }$ and $T_{2}^{\prime }$ be subsets of $T_{1}$ and $T_{2}$ with the property that their elements are closer to the mass centers ${\bar{t}}_{1}$ and ${\bar{t}}_{2}$ compared to $T_{1}$\$T_{1}^{\prime }$ and $T_{2}$\$T_{2}^{\prime }$ (the symbol “\” is used for exclusion). In order to find the new subsets $T_{1}^{\prime }$ and $T_{2}^{\prime }$ a proper threshold would be set to 90%; only 90% of the elements of $T_{1}$ and $T_{2}$, closest to ${\bar{t}}_{1}$ and ${\bar{t}}_{2}$, are constructing the subsets $T_{1}^{\prime }$ and $T_{2}^{\prime }$. Now, ${\bar{t}}_{1}^{\prime }$ and ${\bar{t}}_{2}^{\prime }$ are defined as the mass centers of $T_{1}^{\prime }$ and $T_{2}^{\prime }$. The third distance function (fulfilling the distance properties) excluding the outliers is:$$d_{3}\left(T_{1},T_{2}\right)=\rho \left({\bar{t}}_{1}^{\prime },{\bar{t}}_{2}^{\prime }\right)$$

The three different distance functions proposed above have some significant differences. The first function $d_{1}$ describes the distance between two sets of microsaccades using the best case, i.e., it can be considered as an optimistic estimate of the distance between two segments. The second function $d_{2}$ relates the determination of the distance between two segments to the distance between the centers of two clusters of microsaccades. From a statistical point of view, this is the most plausible solution. The last assessment $d_{3}$ brings an interesting and very useful nuance—combating outliers. Like many other methods in statistics using ranking, the proposed method discards the worst 10% of microsaccades and uses the rest to calculate the cluster centers and their associated distances. Although the latter method is the most computationally intensive, it will provide the most reliable estimate of proximity in the presence of outliers.

#### 3.4.4. Distance between Sets of Segments (Persons)

An approximation of the distance between two persons $K_{1}$ and $K_{2}$ is defined analogically:$$d_{P}\left(K_{1},K_{2}\right)=\underset{T_{1}\in K_{1},T_{2}\in K_{2}}{min}\left(d\left(T_{1},T_{2}\right)\right),$$
where $K_{1}$ and $K_{2}$ are sets with 5 segments in each, corresponding to the fixative points of the compared two persons.

When selecting a distance function, a question may arise as to why the minimum distance between two microsaccades/two segments/several fixation points is chosen as the distance. The logic of choosing the “optimistic” distance estimate, as we called it earlier, is as follows. The distance is determined by the closest elements (microsaccades, segments or people) in the respective clusters. Here, proximity means the proximity of the parameters characterizing the corresponding element of the cluster. Let us assume for a moment that we take the maximum distance to determine the distance. On one hand, this is a stricter condition, which will cause greater proximity of all the elements of the clusters. On the other hand, this distance will usually be determined by abnormal realizations or outliers. Obtaining estimates of samples with outliers does not lead to a good final result in random process statistics.

#### 3.4.5. Error Function

In this section an error function is introduced. The value of this function for a selected vector of weights will give us a quantitative estimate of the choice made. A smaller value of the error function indicates a better choice of the weights vector:$$Err\left(w\right)=\frac{\sum_{i=1}^{N}\sum_{1\leq j_{1}<j_{2}\leq 5}^{5}d\left(K_{i}^{j_{1}},K_{i}^{j_{2}}\right)}{\sum_{i=1}^{N}\sum_{k=1}^{N}\sum_{1\leq j_{1}\leq 5}^{5}\sum_{1\leq j_{2}\leq 5}^{5}d\left(K_{i}^{j_{1}},K_{k}^{j_{2}}\right)},$$
where $K_{i}^{j}$ is the i -th person of the j -th segment, N is the number of people in the experiment, and w is the weight vector. The number 5 refers to the fact that 5 fixative points are given in the scenario of the experiment. The definition of the error represents the fraction of the following two expressions:-The sum of all distances between all segments of a single person (numerator);-The sum of the distances between the segments of every two different persons (denominator).

In Figure 4, the most important definitions for the algorithm realization are shown:

### 3.5. Computation of the Weights

To determine the distance between two microsaccades, two segments (each consisting of several microsaccades), or two individuals (each with several recorded segments), weights must be calculated in advance. The minimizer of the error function guarantees that the difference between two segments of the same person is relatively low and the difference between two segments of different persons is relatively high. Given the error function, the weights vector $w={\left\{w_{i}\right\}}_{i=1}^{N}$ should be found as a solution of the problem:$$w=\underset{v\in R^{N}}{argmin}\left(Err\left(v\right)\right)$$

The algorithm for solving the optimization problem is described in Figure 5.

#### 3.5.1. Problem Properties

The error function by definition is represented as a fraction of two polynomials. In it, the coefficients of the weight vector are the variables. This function has the following property: the value of the error function will not change if we multiply all components of the vector of weights by the same positive, real coefficient:$$Err\left(\alpha w\right)=Err\left(w\right)$$

This property transforms the original problem into the following:$$w=\underset{\left|v\right|=1,v\in {\left(R^{+}\right)}^{N}}{argmin}\left(Err\left(v\right)\right)$$

All components of the weight vector are positive.

#### 3.5.2. Linear Search

The steepest descent algorithm is used for solving the problem. The start point $w_{0}$ is chosen by a random vector generator. It is a normalized random vector (N uniformly distributed variables, which are then normalized to a unity vector). On every step, the gradient $\nabla Err\left(w_{i}\right)$ is computed. The direction of movement is chosen as opposite to the gradient: $dir=-\nabla Err\left(w_{i}\right)$. The new point $w_{i+1}$ is chosen as: $w_{i+1}=w_{i}+k.dir$, where k is the largest number in the sequence 1, 1/2, 1/4, 1/8, etc., so that:$$Err\left(w_{i}+k.dir\right)<Err\left(w_{i}\right)$$
holds true. The sequence $w={\left\{w_{i}\right\}}_{i=0}^{\infty }$ is constructed consequently with this procedure. The stop criterion is chosen when the pseudo $error=\|w_{i}-w_{i+1}\|$ becomes lower than a predefined threshold. It should be noted that the posterior error is different from the $Err\left(.\right)$ function. The main purpose of the algorithm is to successfully identify a very high percentage of the participants. The posterior error is defined as the reciprocal of the number of unsuccessfully identified people.

If at some point a component of the weight vector is found to be less than zero, it is zeroed. This ensures that the solution remains in the definite area.

#### 3.5.3. Improving the Method

The linear search case described above is easily solved by the method of steepest descent. The found minimum of the error function also uniquely determines the vector of weights with a length of 13 elements. The problem is that the adopted linear model is too simplistic and does not allow for the description of more complex relationships. A quasilinear approach is a suitable solution in this case. The main problem associated with its application is the increased computational load when solving the optimization task. The number of components of the weight vectors is dramatically increased—from 13 to 91. When calculating the gradient, the method needs significant computer resources.

A problem arises in the implementations of the linear and quasilinear algorithms, which should not be ignored. When performing steps in the opposite direction of the gradient, one often goes out of the definition area, and the corresponding coefficients are reset to zero. It is possible that in some cases only a few coefficients remain non-zero. This is unacceptable, and special algorithmic measures are taken to stay within the definitional area. In summary, the following two features of the proposed algorithm (which are actually common to this class of algorithms) should be taken into account:-Gradient calculation should be performed as rarely as possible because it requires a large amount of calculation time;-Steps in a given direction must be precisely controlled to stay within the definition area or near the local extremum.

#### 3.5.4. Description of the Minimum Search Algorithm

A start point $w_{0}$ is randomly chosen. The gradient is computed at the point $w_{i}$, and the direction is chosen as $dir=-\nabla Err\left(w_{i}\right)$. A small step ξ is chosen. While $Err\left(w_{i}+l\xi \right)<Err\left(w_{i}\right)$, l is crawling the natural numbers. When the last l is found (for which the last inequality holds true), one additional step is done in the same direction. Then, a new gradient and step are computed with the simple inculcation that the step should not exceed ξ.

The algorithm is executed 30 times with 30 different random starting points $w_{0}$. The output $w^{*}$ corresponds to the smallest value of the function *Err*(.). The first 13 values of $w^{*}$ are the weights of the parameters from Section 5 in the same order as they were described there: the first value corresponds to the weight of the duration, the second to the height, the third to the area, etc. The remaining 78 values correspond to the quasilinear model: all combinations between two parameters are again in the same order as in the description given in Section 5. The 14th parameter is corresponding to the combination duration–duration, the 15th parameter is corresponding to the combination duration–height, the 16th parameter is corresponding to the quasilinear combination duration–area, etc. There are some similarities in the obtained weight vectors. Most of the first 13 parameters are zeroes. This implies that relations between the pairs of parameters are more important for solving the problem, compared to only single parameters. The calculated weight vector is shown in Appendix A.

The large amount of zeroes observed above indicates that some parameters or quasilinear combinations of parameters are irrelevant for the solved task.

#### 3.5.5. Computational Complexity

The algorithm with eye tracker data for 4 persons (everyone is described by 5 segments, consisting of 15,000 recordings for each segment) is run on an Intel(R) Core(TM) i7-7700 CPU, 3.60 GHz, 4 CPUs, 8 threads. The computation time (for all 30 start points) is about 15 h. On average, this means there are about 30 min of computation time needed for the calculation of one local minimum. We found that the gradient calculation required the most computing resources. The algorithm uses adaptive step selection to minimize frequent gradient computation. Calculating the error function requires relatively less computer resources, proportional to the square of the number of microsaccades compared.

## 4. Experimental Design

### 4.1. Experimental Setup

An eye-tracking device ”Jazz novo” (Ober Consuting Sp. Z o.o.) is used in the observations. The recording rate of the eye tracker (ET) is 1 kHz. This mobile eye-tracking device measures several types of data:-Integral (monocular) data about horizontal eye position in degree (horizontal eye tracker);-Integral (monocular) data about vertical eye position in degree (vertical eye tracker);-Head rotation velocity Y (pitch) in deg/s (often this sensor is called gyroscope/gyro, not very correctly);-Head rotation velocity Z (yaw) in deg/s;-Head acceleration—horizontal in g (horizontal according to the head accelerometer);-Head acceleration—vertical in g (vertical according to the head accelerometer).

“Jazz novo” is regarded as a mobile, non-obtrusive device fixed above the person’s nose.

The observed scenario is carefully prepared in advance on PowerPoint video, strictly maintaining the timing of the serial events. This scenario is presented on an NEC MultiSync LCD monitor with a mean luminance of 50 cd/m^2^ and a resolution of 1280 × 1024 pixels. To minimize visualization delay, a computer with an Nvidia Quadro 900XGL graphics board is used. The equipment is located in a darkened room in which only the test subject is present during the experiment at a distance of 58 cm from the display screen. The given distance to the screen was chosen for correspondence between an eye rotation of 1 degree and a 1 cm fixation point shift on the screen (approximately).

Eight healthy (in their opinion) persons were examined. Six of them were of ages between 20 and 30 years (students and young researchers). The other two were older than 40 years. All participants provided written informed consent for participation in a 5′ test (the real one was shorter).

### 4.2. Test Scenario

The test scenario is designed with the help of Microsoft PowerPoint animation. Several visual tasks were realized: fixation of gaze on a fixed point, tracking of a rectilinearly moving object in horizontal and vertical directions, a task of searching for an object in a complex picture, a task of counting objects and a task of searching for an error in a text. In this article, only the results of the first experiment are discussed. It includes a fixed bright dot with a diameter of 1 cm appearing in the middle of the left edge of the screen, on the middle of the right edge, on the middle of the top edge of the screen, on the middle of the bottom edge and finally in the middle of the screen. The fixed points are presented for exactly 15 s. The recording subject was sitting alone in the experimental room, and the room was dimmed to minimize visual distractions.

## 5. Results and Discussion

The derived weights in the last section are applied to the person identification problem. The distance between persons’ segments is used to estimate the level of differentiation between the different persons. A higher distance between two persons means a higher difference between the parameters of their microsaccade trajectories, which leads to lower chance of them being detected as the same person. A lower distance between two persons analogically determines a higher chance of them being observed as the same person during different periods of time. The distances between segments are defined as $D_{i_{1},i_{2}}^{j_{1},j_{2}}=d\left(T_{i_{1}}^{j_{1}},T_{i_{2}}^{j_{2}}\right)$, where $T_{i}^{j}$ is the jth segment of the ith person. For every couple of persons, the distances $D_{i_{1},i_{2}}^{j_{1},j_{2}}$ are summed and averaged. The obtained values:$$F_{i,k}=\frac{1}{25}\overset{5}{\underset{j_{1}=1}{\sum }}\overset{5}{\underset{j_{2}=1}{\sum }}D_{i,k}^{j_{1},j_{2}}$$
are the distances between every two persons i and k.

### 5.1. Person Identification

For the purpose of automatic person identification, the parameters’ data are preprocessed using unit vector normalization. The data set are split into training and test data. For each person, only four fixative points (left, right, down and center) are used for training the model (training space). The remaining fifth fixative point (up) is used as test data. The level of correct prediction of which person is the correct owner of the fifth fixative point is an estimate of the quality of the algorithm.

### 5.2. The Test Procedure

The weights are computed with the described algorithm, using only four fixative points. The distances between every two microsaccades could be computed through the weight vector. This provides a distance between every two segments. In order to test the algorithm, we are interested in the distances between the following two objects:

-A person, represented by their four segments (the segments in the training space);-A segment from the test space.

The above is achieved by summing the distances corresponding to each of the four segments of a person. The result is the distances between individuals (a set of four segments) and the fifth segment (corresponding to the fifth fixative point). This is summarized in Table 1.

The four segments of each person are a subset of the training space, and the fifth segments are elements of the test space. No test data are used for training (weight calculation), but the training data participate in distance calculation for test data.

From the cells of Table 1, the cell with the smallest value is chosen. The corresponding pair Segment 3–Person 3 is found. The corresponding row and column are discarded, and the same procedure is repeated again with the remaining 3 × 3 table: L1 normalization of the rows and consequent choice of the smallest value. The procedure is repeated until all segments are related to the corresponding persons.

The described algorithm is run again over 30 random starting points $w_{0}$. All of the four persons were related with the correct missing segment for each of the 30 different local minima w, corresponding to all of the 30 random starting points.

### 5.3. Interpretation of the Computed Weights

From the set of all 30 local minima, one is chosen with the lowest Err value ($w^{*}$), which is considered as a global minimum. Its parameters’ weights are used for analysis of the importance of each parameter. The components of the computed $w^{*}$ vector of weights, rounded to the fifth sign, are shown in Appendix B.

Higher values of the components of $w^{*}$ correspond to higher importance of the respective parameters. The first 13 values are the weights of the above-described 13 parameters. The most important parameter for the purpose of person identification is the average microsaccade speed, with a weight value of 0.01694. The second one is the height of the microsaccade, with a weight of 0.01484. The third one is the maximal microsaccade diameter, with a weight value of 0.01358. The fourth one is the microsaccade base length, with a weight value of 0.01129. The remaining nine parameters have weight values lower than or equal to half of the weight value of the parameter “height”. Two of them have no impact on person identification because their weight values are equal to 0 (duration and sharpness).

By analogy, an arrangement of the importance of the combinations of parameters could be constructed for the remaining 78 components of $w^{*}$. The most important among them is the combination microsaccade duration × sharpness, with a weight value of 0.98345. This explains why the quasilinear model is much more precise than the linear model: the parameter sharpness is irrelevant by itself, with a value 0, but in a combination with the duration it becomes the most significant indicator for person identification. The reason for the value of sharpness being 0 is hidden in its correlation with the parameters height and area. These two parameters (height and area) are used in the algorithm, and their combination does not yield additional information in the linear part. However, in the quasilinear part the combination of height, area^−1^ and duration becomes a very important feature. It is in fact a nonlinear combination because three parameters are used, and one of them is raised to the power of −1. This is the motivation for using correlated parameters: the power of the nonlinear case is used partly in the quasilinear case without a loss of computational power.

More detailed results about the importance of the parameters can be found in Appendix A and Appendix B, where the optimal weight vector is given for five fixative points (Appendix A) and for four fixative points (Appendix B). However, the results from Appendix A and Appendix B have many similarities.

In the second case (four fixative points) the corresponding error value to $w^{*}$ is 0.11741. The vector corresponding to the errors of the 30 local minima is:ErrorsOnLocalMinimums = [0.1478, 0.1475, 0.14746, 0.14792, 0.14768, 0.14715, 0.14792, 0.14687, 0.14706, 0.1457, 0.14608, 0.14604, 0.14787, 0.14441, 0.14796, 0.14786, 0.14715, **0.11741**, 0.14769, 0.14556, 0.14338, 0.14597, 0.14797, 0.14495, 0.14761, 0.14424, 0.14797, 0.14698, 0.14771, 0.14709],
where the minimum value is written in boldface.

The errors are drastically decreased compared to the errors at the 30 starting points, as shown in:ErrorsOnStartingPoints = [0.22448, 0.19845, 0.19674, 0.27169, 0.22612, 0.18583, 0.24529, 0.17309, 0.21237, 0.23558, 0.17295, 0.16734, 0.26249, 0.19341, 0.29925, 0.24706, 0.18597, **0.1554**, 0.2046, 0.21945, 0.19088, 0.24447, 0.33729, 0.20171, 0.22902, 0.20583, 0.29074, 0.17101, 0.22931, 0.19944].

Here, the point corresponding to the last minimum is written in boldface.

### 5.4. Interpretation and Application of the Person Identification

The experiments conducted on the pilot sample of four individuals show that all individuals are identified successfully. Furthermore, 80% of the obtained fixation microsaccade data was used for training and the remaining 20% of the data for testing. Initial tests have shown that fixation microsaccades can be successfully used for identification, with potential applications in mobile devices, security gates, digital signatures, etc., with locally embedded eye trackers. A more detailed description of some applications is presented below.

If a smartphone is equipped with proper hardware (precise front cameras) and software (eye detector), an identification eye-tracking system could be developed on the base of the described algorithm. The smartphone could be unlocked with a fixative gaze to the camera. This would be more secure compared to facial recognition identification systems because no photo or 3D mask could be used for adulteration of the facial recognition system. On security gates the eye tracker could be used for a higher level of reliability.

The eye-tracking identification method could be also combined with other, already well-known identification methods. A digital signature system easily could be combined with personal identification by an eye-tracking system. In our digital world, it is a great priority to identify a person by biometrical data. Most known signatures own a material component. For example, hand signatures have a material component on sheets of paper. A fingerprint has also a material component on the finger of each person. Most passwords are easily broken with a brute force attack, or if the passwords are stronger, normally they are stored again on a material component—a computer or a sheet of paper. Eye-tracking based identification does not need a material component. This fact gives this system a great advantage over other ones.

## 6. Conclusions and Future Work

This article analyzes small eye movements in fixation mode in a pilot study. A scenario of gaze fixation on a point was chosen so that the specific cognitive process did not influence these movements. The specificity of tremors, drifts and microsaccades was examined, and microsaccades were selected for identification purposes. They have easily distinguishable characteristics and a larger amplitude, which allows their use with a less precise ET. In order to perform the identification, a parameterization of the microsaccades is performed. Thirteen features were selected for further analysis. The optimal weights are calculated by using gradient algorithm. The obtained initial results for person identification show that, although the pilot sample is small, microsaccades could be used for biometric needs.

The present work represents only the first step in proving the concept of using microsaccades in biometrics. For the engineering implementation of such a system, it is necessary to investigate and solve many other problems. The most important of them are the following. It is necessary to conduct research on the influence of a person’s mental state on the statistical characteristics of microsaccade parameters; the influence of some chemical substances on them; and what the correlation is between the parameters of microsaccades and the general state of health of a person, his age, gender, race and in particular the state of the eyes. An important factor in biometrics is how much the statistical characteristics of a person’s parameters remain constant over a month, a year or a decade. The studies described above should also be implemented using a group of participants at least two orders of magnitude larger than the sample size of the pilot study. Only in such a group will it be possible to prove the discriminative capabilities of the algorithms. Current research shows that a natural development of the algorithm is its implementation using neural networks. Much more can be done in selecting appropriate distance functions. Building specific distance functions would enable physicians to use the system for diagnostic needs in medical practice.

## Figures and Tables

**Figure 1 sensors-23-00089-f001:**
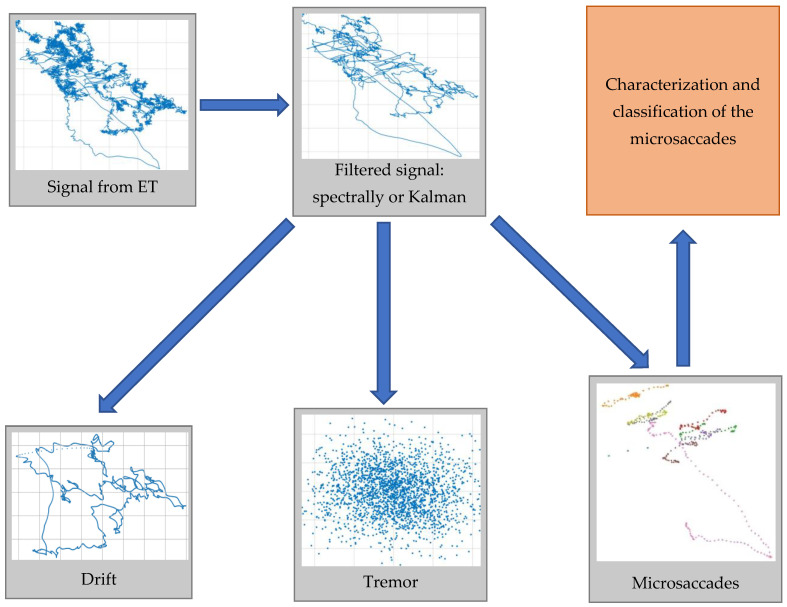
The preprocessing steps of filtration of the data and separation of the three different fixative eye movements.

**Figure 2 sensors-23-00089-f002:**
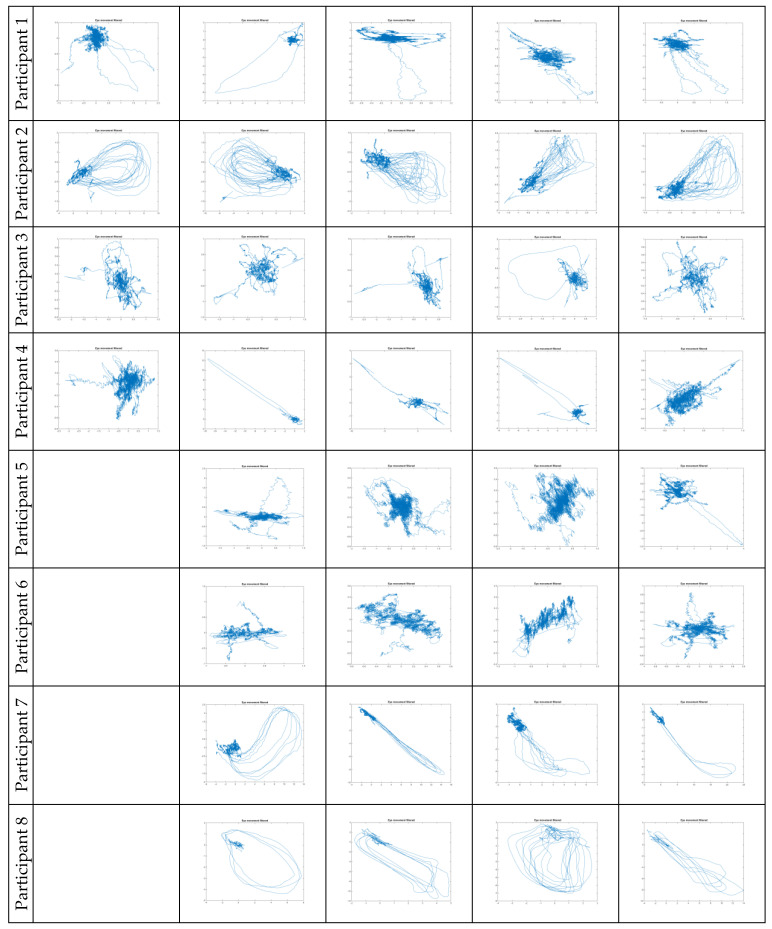
The fixative eye movements for left, right, top, bottom and center points, respectively, for each of the participants.

**Figure 3 sensors-23-00089-f003:**
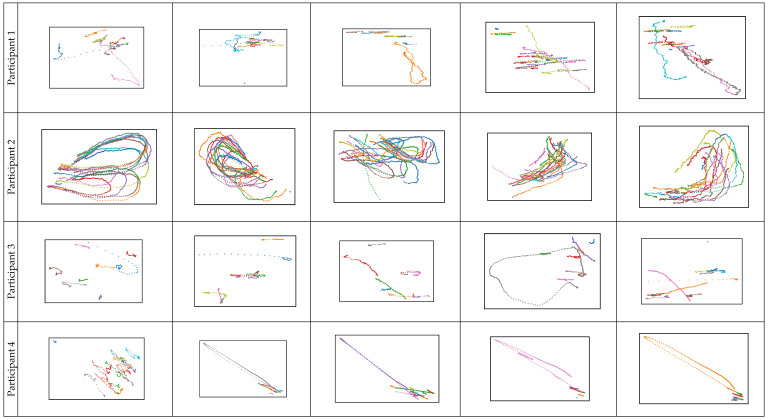
The microsaccades only for left, right, top, bottom and center points, respectively, for the first four of the participants.

**Figure 4 sensors-23-00089-f004:**
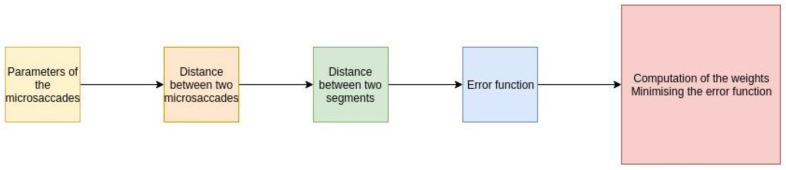
A set of definitions needed to implement the algorithm.

**Figure 5 sensors-23-00089-f005:**
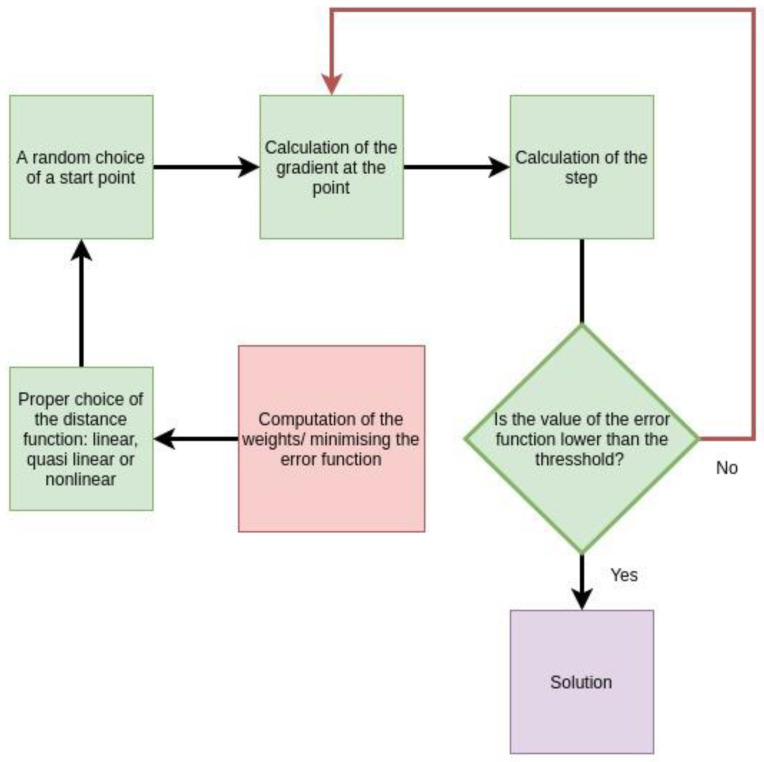
The main steps of the algorithm.

**Table 1 sensors-23-00089-t001:** The total distances between the four segments of every person and the fifth segment in the test space.

	Person 1	Person 2	Person 3	Person 4
Segment 1	0.102	0.613	0.108	0.177
Segment 2	0.186	0.333	0.229	0.252
Segment 3	0.0624	0.769	0.023	0.146
Segment 4	0.172	0.434	0.191	0.203

## Data Availability

Data from experiments are available upon request from e-mail addresses of the authors of the article.

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
