# Peer review of "Can Microsaccades Be Used for Biometrics?"

_sensors, 2022, doi:10.3390/s23010089_

Round 1

Reviewer 1 Report

The paper asks the question of whether microsaccades can be used as an alternative biometric trait or not. Overall the work is original and interesting. However, the presentation requires significant improvement. 

Some specific comments:

1. The author must compare their traits with any other biometric techniques and state that how unique is the proposed one in comparison to others. It's unclear if we can uniquely repeat the experiment hence reproducibility is one of the main concerns. However, I believe with adding more subjects one can asses the feasibility of the approach. 

2. The eye-tracking device may not be easily accessible to a variety of users. Hence, how it can be used for day-to-day usage is unclear. Moreover, the application aspects of the used method are not very clear in the paper. The author should clearly state the application areas that they are targeting. 

3. The entire procedure should be explained using the pseudocode or flow diagram. The explanation of the approach needs to be strengthened. 

4. Most of the references are outdated. I don't see the author citing the recent work in their paper. 

5. Interpretation of weight is unclear to me. It is a good idea to move the large weight vector as an appendix and refer to them in the running text. Right now it is very discrete and unorganized. 

6. The number of experiments is very few and it's unclear how easy it is to add more subjects for a better comparison. 

7. The details of control experiments and targetted experiments are not apparent and require improvement. 

8. Why 13 odd features are required. are these features correlated?  Can we apply dimensionality reduction? 

9. The time complexity of one experiment etc is also needed. how many minimum lengths of samples are needed for a fair comparison? 

Author Response

  1. The author must compare their traits with any other biometric techniques and state that how unique is the proposed one in comparison to others. It's unclear if we can uniquely repeat the experiment hence reproducibility is one of the main concerns. However, I believe with adding more subjects one can assess the feasibility of the approach.

Answer: A new chapter 2 “Related work” is added. It is explained that no other methods for biometrics through fixative eye movement are made.

  1. The eye-tracking device may not be easily accessible to a variety of users. Hence, how it can be used for day-to-day usage is unclear. Moreover, the application aspects of the used method are not very clear in the paper. The author should clearly state the application areas that they are targeting.

Answer: The section “Interpretation and application of the person identification” has been extended with a more detailed application for the wide users. Also smartphone cameras are suggested as potential eye trackers. The areas of application are described in the new clearer context of application- smartphones, security gates and digital signatures.

  1. The entire procedure should be explained using the pseudocode or flow diagram. The explanation of the approach needs to be strengthened.

Answer: Two new diagrams are added- one with the new concepts, used in the algorithm and another containing all main steps.

  1. Most of the references are outdated. I don't see the author citing the recent work in their paper.

Answer: The number of references was 34, from which 14 since 2014 till 2022. New 22 references were added, fourteen of which are published in the last 10 years, four in the last 2 years. The library is updated with more new articles, which are cited mainly in the new chapter “Related work”

  1. Interpretation of weight is unclear to me. It is a good idea to move the large weight vector as an appendix and refer to them in the running text. Right now it is very discrete and unorganized.

Answer: A more detailed interpretation of the weight vector is added in chapter “Interpretation of the computed weights”. The two long vectors of the weights are moved respectively in Appendix 1 and Appendix 2.

  1. The number of experiments is very few and it's unclear how easy it is to add more subjects for a better comparison.

Answer: The methodology for adding high number of new people is discussed in the new Chapter “Conclusions and future work”. As would be stated in the answer of remark 13 the computational complexity for high number of persons is newly discussed in Section “Description of the minima search algorithm”

  1. The details of control experiments and targeted experiments are not apparent and require improvement.

Answer: The data set is split to training and test data. For each person only four fixative points (left, right, down and center) are used for training the model (training space). The remaining fifth fixative points (up) are used as test data. (Section 4.3.1 Person identification)

  1. Why 13 odd features are required. are these features correlated? Can we apply dimensionality reduction?

Answer: In order to describe a microsaccade in details are required at least 10 different features. Every number above it (of uncorellated features) would be sufficient for the needs of person identification. Some of the parameters are correlated with other (for example sharpness is correlated with the height and area-1). The reason why such parameters are kept is that in the quasilinear model they bring higher degree of preciseness. The importance of the parameter sharpness is newly explained in section “Interpretation of the computed weights”. The sharpness is irrelevant by itself (because it is correlated with other two parameters), but in a combination with the duration becomes the most important parameter.   

  1. The time complexity of one experiment etc is also needed. how many minimum lengths of samples are needed for a fair comparison?

Answer: A new chapter 3.5.5. “Computational complexity” is added. We recommend that the number of samples is not less than 10000 per (corresponding to a 10 sec. record with frequency 1000 Hz) in order to have rich enough microsaccades set.

Reviewer 2 Report

The content of this paper is to introduce the novel technique that microsaccades to be used for biometrics. It provided useful information and propose some new theories. However, the content needs to be enhanced before it could be recognized as a research paper.

1.The authors should do more experiments More data are required to analyze. Only eight persons were tested and 6 people are in the young stage. What is the difference between young and old people? What is the difference between males and females? Please perform more experiments for more age ranges of people.

2.Please write the paper and divide it into “Material and Methods” and “Results”. A section of theoretical analysis could be used if necessary.

3.The data in 8.3 could be moved to the section of the Appendix.

 After rewriting this paper, please resubmit it.

Author Response

  1. The authors should do more experiments. More data are required to analyze. Only eight persons were tested and 6 people are in the young stage. What is the difference between young and old people? What is the difference between males and females? Please perform more experiments for more age ranges of people.

Answer: We have observed a strong difference between males and females sight. The dispersion in male fixative movement is much lower than the dispersion in female fixative movement. The male fixation is “more concentrated” than the female fixation. Unfortunately we have not observed any marks in the microsaccades for classification of persons by their gender.

About the age classification we have observed that the microsaccades area and microsaccades number is increasing with the age. Currently we have no enough data to classify the age. It would be a good subject for another article combining age and gender characteristics with other personal indicators. 

  1. Please write the paper and divide it into “Material and Methods” and “Results”. A section of theoretical analysis could be used if necessary.

Answer: The article is fundamentally reorganized as shown below: 

Old structure

New structure

1. Introduction

1. Introduction

2. Test equipment

2. Related works

3. ET data preprocessing

3. Proposed method

4. Motivation for microsaccade analysis and microsaccade parameterization

3.1. ET data preprocessing

5. Parameterization of the microsaccades

3.2. Motivation for microsaccade analysis and microsaccade parameterization

6. Distance functions

3.3. Parameterization of the microsaccades

6.1. Linear distance function

3.4. Distance functions

6.2. Quasilinear distance function

3.5. Computation of the weights

6.3. Segment distance

4. Experimental design and results

6.4. Distance between sets of segments (persons)

4.1. Test equipment

6.5. Error function

4.2. Test scenario

7. Computation of the weights

4.3. Description of the algorithm

7.1. Problem properties

4.4. Results

7.2. Linear search

5. Conclusions and future work

7.3. Improving the method

7.4. Description of the algorithm

8. Results

8.1. Person identification

8.2. The test procedure

8.3. Interpretation of the computed weights

8.4. Interpretation and application of the person identification

9. Conclusion

  1. The data in 8.3 could be moved to the section of the Appendix.

Answer: The two big weight vectors are moved in the new Appendix sections.

Round 2

Reviewer 1 Report

the paper can be acceptted 

Author Response

English language and style were checked and many inaccuracies were corrected.

Reviewer 2 Report

The content of the revised version has improved significantly. 

Section of 4 Experimental design and results should divide into 4. Materials and Methods and 5. Results and discussion.

The section of the 5. Conclusion and future work should revise as 6. Conclusion.

Author Response

  1. English language and style were checked and many inaccuracies were corrected.
  2. Section 4 is divided into  

        4. Experimental design

        5. Results and discussion.

  3.  

    Section 5. Conclusions and future work is renumbered to 6.
